



# Modelled forest ecosystem carbon-nitrogen dynamics with integrated mycorrhizal processes under elevated $CO_2$

Melanie A. Thurner[1,2,3], Silvia Caldararu[1], Jan Engel[1], Anja Rammig[3], and Sönke Zaehle[1,4]

[1]Max Planck Institute for Biogeochemistry, Jena, Germany
[2]International Max Planck Research School for Global Biogeochemical Cycles, Jena, Germany
[3]TUM School of Life Sciences Weihenstephan, Freising, Germany
[4]Michael Stifel Center Jena for Data-driven and Simulation Science, Jena, Germany

**Correspondence:** Melanie A. Thurner, now at Universität Hamburg (melanie.thurner@uni-hamburg.de)

**Abstract.** Almost 95% of all terrestrial plant species form symbioses with mycorrhizal fungi that mediate plant-soil interactions: Mycorrhizae facilitate plant nitrogen (N) acquisition and are therefore vital for plant growth, but also build a pathway for plant-assimilated carbon (C) into the rhizosphere. Therefore, mycorrhizae likely play an important role in shaping the response of ecosystems to environmental changes such as rising atmospheric carbon dioxide ($CO_2$) concentrations, which can increase

plant N demand and the transfer of plant C assimilation to the soil. While the importance of mycorrhizal fungi is widely recognised, they are rarely represented in current terrestrial biosphere models (TBMs) explicitly. Here we present a novel, dynamic plant-mycorrhiza-soil model as part of the TBM QUINCY. This new model is based on mycorrhizal functional types that either actively mine soil organic matter (SOM) for N or enhance soil microbial activity though increased transfer of labile C into the rhizosphere and thereby (passively) prime SOM decomposition. Using the Duke Free-Air $CO_2$ Enrichment (FACE) experi-

ment, we show that mycorrhizal fungi can have important effects on projected SOM turnover and plant nutrition under ambient as well as elevated $CO_2$ treatments. Specifically, we find that including enhanced active mining of SOM for N in the model allows to more closely match the observations with respect to observed decadal responses of plant growth, plant N acquisition, and soil C dynamics to elevated $CO_2$, whereas a simple enhancement of SOM turnover by increased below-ground C transfer of mycorrhizae is unable to replicate the observed responses. We provide an extensive parameter uncertainty study to investi-

gate the robustness of our findings with respect to model parameters that cannot readily be constrained by observations. Our study points to the importance of implementing mycorrhizal functionalities in TBMs as well as to further observational needs to better constrain mycorrhizal models and to close the existing major knowledge gaps of actual mycorrhizal functioning.

## 1 Introduction

The land biosphere sequesters about a quarter of the annual anthropogenic $CO_2$ emissions and thereby slows human-induced climate change (LeQuéré et al., 2018). However, it is unclear if this buffering effect will continue into the future (Ciais et al.,





2014). Multiple lines of evidence (Norby et al., 2005; Jiang et al., 2020; Walker et al., 2020) and the majority of nitrogen-enabled terrestrial biosphere models (TBMs; Zaehle et al., 2014; Wieder et al., 2015; Fleischer et al., 2019) suggest that $CO_2$-fertilised plant growth, which is one of the major causes for this uptake, will be reduced by increasing nutrient limitation. Whole ecosystem experiments, in which plants are exposed to elevated levels of atmospheric $CO_2$ (e$CO_2$), show a range

of responses from on-going $CO_2$-fertilised growth (McCarthy et al., 2007) to progressive nutrient limitation (Norby, 2010). One possible explanation for these divergent responses is the differential ability of plants to accesses soil nutrients through mycorrhizal fungi (Terrer et al., 2016).

Mycorrhizal fungi are ubiquitous and form symbioses with almost 95% of all terrestrial plant species (Read (1991)). It is commonly assumed that this mutualistic relationship between plants and fungi is an evolutionary adaptation to nutrient-poor

conditions (Read, 1991), where host plants benefit from mycorrhizal nutrient supply (Ames et al., 1983; Marschner and Dell, 1994; Ek et al., 1997). In return, plants export carbohydrates to hosted fungi, which cover most of fungal carbon (C) demand (Gorka et al., 2019). Plant-mycorrhiza symbioses are important for soil C sequestration and thus soil organic matter (SOM) content, either by allocating C into below-ground tissues such as roots or mycorrhizal biomass, or by rhizodeposition, i.e. C exudation into the rhizosphere of the soil (Godbold et al., 2006; Frey, 2019; Gorka et al., 2019).

Mycorrhizae are generally classified based on their morphology into arbuscular mycorrhizae, ecto-mycorrhizae, ericoid, and orchid mycorrhizae, but the first two are the most common forms (Read, 1991; Johnson and Jansa, 2017). Arbuscular mycorrhizae (AMs) directly enter plant root cells and build symbiotic associations with more than 80% of all terrestrial plant families, especially with crop and grass species, but also with most tropical trees, which is why they are often assumed to be more important for plant phosphorus (P) nutrition (Read, 1991; Hodge et al., 2001). Ecto-mycorrhizae (ECMs) are hosted by

only 10% of plant species that include most temperate and boreal tree species. ECMs are usually regarded as nitrogen (N) nutrition supporters, and only cover plant roots, working as a storage and exchange area for carbohydrates and nutrients, but also as protection against infections and as a physical barrier between roots and soil nutrients (Read, 1991; Jordy et al., 1998; Laczko et al., 2004; Pritsch et al., 2004; Finlay, 2008).

Given the vast diversity of mycorrhiza species, even after decades of research, there is still no complete understanding on

how plant-mycorrhiza symbioses affect ecosystem nutrition or the stabilisation and turnover of soil organic matter (SOM) (Jansa and Treseder, 2017; Tedersoo and Bahram, 2019). Some species accelerate SOM decay by C exudation to soil, which activates the microbial community and thus enhances SOM decomposition rates (Hodge et al., 2001; Cheng et al., 2012; Lindahl and Tunlid, 2015; Paterson et al., 2016; Lang et al., 2020). Other mycorrhizal species have the ability to take up nutrients directly from SOM in addition to mineral nutrients (Wu et al., 2005; Pérez-Tienda, 2012; Hodge and Storer, 2015),

or even act saprotrophically, i.e. they are able to gain C from SOM (Treseder et al., 2007; Malcolm et al., 2008; Nehls, 2008), where other species are shown to lack the genetic capacity to act as saprotrophs (Lindahl and Tunlid, 2015; Frey, 2019)).

Because of the knowledge gaps regarding the actual role of mycorrhizal fungi functioning for ecosystems, they are rarely represented in state-of-the-art terrestrial biosphere models (TBMs) that are used to simulate the global carbon and nutrient cycle responses to global change (LeQuéré et al., 2018). Most TBMs predict a higher N limitation effect on $CO_2$-fertilised

growth than observed, potentially due to a significant underestimation of plant N acquisition enhancement under e$CO_2$ (Zaehle





et al., 2014). One possible reason for this is that most TBMs represent N acquisition as plant root uptake of mineral N only, but lack a representation of a plant-controlled pathway to increase N availability and acquisition on demand (Knops et al., 2002; Chapman et al., 2006; Zaehle and Dalmonech, 2011). This could be any pathway, which may involve direct organic nutrient uptake and/or rhizodepostion in order accelerate SOM decomposition, but in particular mycorrhizal mediated rhizosphere-

SOM interactions (Chapman et al., 2006; Phillips et al., 2013; Brzostek et al., 2017). The absence of such pathways can lead to incorrect predictions of plant and ecosystem responses to eCO$_2$, and to uncertain estimates of ecosystem impact on climate change (Brzostek et al., 2017; Terrer et al., 2018; Shi et al., 2019).

Consequently, several approaches to implement mycorrhizal dynamics into ecosystem models have been developed during the last decade either to explore C-nutrient exchange or competition between host plant and mycorrhizal fungi for individuals

(e.g. Meyer et al., 2010; Franklin et al., 2014), or to improve the representation of plant nutrient acquisition in TBMs at the ecosystem scale by simulating their effects on SOM dynamics (e.g. Deckmyn et al., 2011; Orwin et al., 2011), or by including them directly into plant N nutrition calculations (e.g. Phillips et al., 2013; Brzostek et al., 2014, 2015; Sulman et al., 2017). One of the main issues facing such a model is that both species of arbuscular and ecto-mycorrhizal type have been shown to actively mine SOM as a nutrient source, either by a direct enzymatic breakdown of material, which potentially includes saprotrophic

behaviour (Hodge et al., 2001; Wu et al., 2005; Pérez-Tienda, 2012; Hodge and Storer, 2015), or by passively priming SOM by C exudation, which accelerates SOM decomposition (Hodge et al., 2001; Cheng et al., 2012; Lindahl and Tunlid, 2015; Paterson et al., 2016; Lang et al., 2020). Additionally, associations with either AMs or ECMs are identified at species level and it is difficult to aggregate them to the ecosystem level, or appointed to plant functional types (PFTs), which are represented in TBMs. Thus, for the purpose of this study we choose not to implement the common mycorrhizal types, but rather explore what

the consequences of these two mycorrhizal functionalities, i.e. an active SOM mining, saprotrophic mycorrhizal function, and a passive SOM priming mycorrhizal function that accelerate SOM decomposition by C exudation, are on plant N acquisition and ecosystem carbon storage, how explicitly we can model plant-mycorrhiza-soil interactions given the current knowledge and data availability, and finally whether or not this improves the prediction of the observed responses of plant N acquisition at ecosystem scale under ambient and elevated CO$_2$ conditions.

We implement the two alternative representations of mycorrhizal functionalities into the QUINCY model (QUantifying Interactions between terrestrial Nutrient CYcles and the climate system, Thum et al. (2019)), and compare the model predictions for these representations to the standard model, which does not account for mycorrhizal fungi, to asses the robustness of the alternative model structures and their parameterisation. We test all model variants at the Duke Free-air CO$_2$ Enrichment (FACE) site to especially investigate responses to atmospheric CO$_2$ enhancement, and compare model output to observational data,

where available (McCarthy et al. (2006)).



## 2 Methods

### 2.1 QUINCY model

The terrestrial biosphere model QUINCY, is described in detail in Thum et al. (2019). The model simulates photosynthesis and growth of vegetation represented by average individuals of plant functional types, as well as soil biogeochemical processes.

QUINCY represents fully coupled C, N, P and water cycles and traces the isotopic signals of $^{13}$C, $^{14}$C, and $^{15}$N. QUINCY operates on a half-hourly time step to account for short-term acclimation, but many processes have a specific memory time, e.g. days, weeks, months, over which the calculated flux is averaged to also consider longer response times. Below, we briefly present the representation of the most relevant processes for plant N acquisition as well as of soil organic matter dynamics that are affected or modified by the implementation of the mycorrhizal processes.

Plant N acquisition is described as direct uptake of inorganic nitrogen by fine roots. Potential root uptake is assumed to be a function of fine root biomass and mineral soil N content, i.e. ammonium ($NH_4$) and nitrate ($NO_3$). This potential uptake is scaled down in times of low plant N demand, which depends on the ratio of available to required N for growth. Biomass is allocated dynamically above- and below-ground in response to plant nutrient and water demand. Acquired N, as well as assimilated C, are first added to plants labile pool, which is one of two non-structural plant pools represented by QUINCY.

From there C, and N are distributed among all other plant pools, i.e. plants structural pools to represent tissue growth, and plants reserve pool as second non-structural pool for storage, within a week. This allows QUINCY to account for sink and source limitation, and smooths daily variations.

QUINCY includes a representation of soil carbon and nutrient pools, which is based on the CENTURY soil model (Parton et al. (1993)). It accounts for five organic soil pools, i.e. three litter pools, a fast and a slow overturning soil organic material (SOM) pool, and three inorganic soil pools, i.e. $NH_4$, $NO_3$, and $PO_4$. Decomposition is described as first order decay by specific turnover times and pool sizes. Net mineralisation of nutrients occurs if the nutrient immobilisation for litter decomposition is less than the gross nutrient mineralisation from the decomposition of soil organic matter. Plant uptake competes with immobilisation, as well as leaching and nitrification-denitrification related gaseous losses for the available inorganic nutrients.

### 25 2.2 MYC model description

The plant-MYCorrhiza-soil interaction (MYC) model includes plant-mycorrhiza dynamics, and mycorrhiza-soil interactions. Relevant QUINCY fluxes, as well as MYC model fluxes are presented in figure 1.

Plant and mycorrhizae interact through the C flux from host plants to mycorrhizal fungi, the reduction of fine root uptake capacity in the presence of mycorrhizae, as these are assumed to be partially covered by the mycelium and therefore have a

less direct contact with the soil, and the export of nitrogen taken up by mycorrhizae to plant roots. The net C and N acquisition



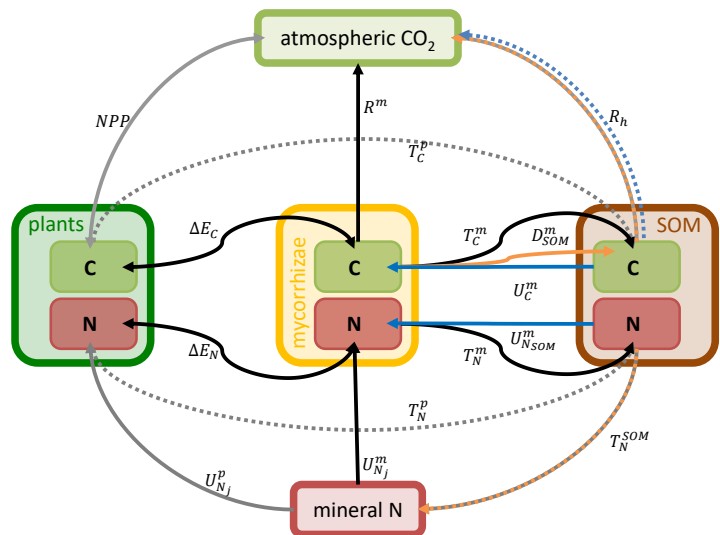

**Figure 1.** Schematic representation of MYC model fluxes within the QUINCY framework. Boxes represent model pools, where light green represents C pools and red represents N pools. They are grouped for plants (dark green), mycorrhizae (yellow) and SOM (brown). Arrows indicate fluxes, where solid arrows are explained in eq. 1 - eq. 12. Dashed arrows are not explained in detail here, as they are not modified by or used within the MYC model. Grey arrows belong to the QUINCY framework, black arrows belong to both mycorrhizal functionalities, blue arrows belong only to saprotrophic mycorrhizae, and orange arrows belong to or are affected by decomposing mycorrhizae. E depicts exchange fluxes, R respiration fluxes, T turnover fluxes, and U uptake fluxes. Subscripts indicate, whether plants (p), mycorrhizae (m), or SOM determine flux rates. Subscripts contain additional information, e.g. exchanged property or source. For more details, see eq. 1 - eq. 12.

of plants (Fig. 2a,b) is described as:

$$\frac{dC^p}{dt} = BP = NPP + \Delta E_C \tag{1a}$$

$$\frac{dN^p}{dt} = A^p_N = U^p_N + \Delta E_N \tag{1b}$$

The net C acquisition of plants, i.e. plant biomass production (BP), is given by net primary production (NPP) and net C exchange between host plant and mycorrhizal fungi ($\Delta E_C$; fig. 2a). Plant (net) N acquisition ($A^p_N$) is given by direct root uptake ($U^p_N$) and net N exchange ($\Delta E_N$; fig. 2b).

NPP and root N uptake are calculated by the following in the standard QUINCY model, and further modified to account





for the effects of mycorrhizae.

$$U_{N,j}^{p,pot,*} = v_{max,j}^* \times N_j \times (K_{m1,j} + \frac{1}{K_{m2,j} + N_j}) \times \zeta_N^p \times C_{fr}^{p,*} \qquad , where \tag{2a}$$

$$v_{max,j}^* = \frac{v_{max,j}}{1 - f_{cover,max} + f_{cover,max} \times f_{U_N^{eff}}^m} \tag{2b}$$

$$C_{fr}^{p,*} = C_{fr}^p \times f_{cover} \qquad , where \tag{2c}$$

$$f_{cover} = MIN(f_{cover,max}, \frac{C^m}{C_{fr}^p}) \tag{2d}$$

where j as index refers to either $NH_4$, or $NO_3$, and $N_j$ denotes $NH_4$, or $NO_3$ concentration in soil. $v_{max,N_j}^*$ is a parameter describing the maximum uptake rate per unit fine root biomass ($C_{fr}^{p,*}$), and $K_{m1,N_j}$ and $K_{m2,N_j}$ are the low- and high-affinity half-saturation parameters. $\zeta_N^p$ is a scaling factor, which describes current plant N demand. The modification of the maximum uptake rate per unit biomass ($v_{max,j}^*$) maintains the overall maximum uptake capacity of the rhizosphere ($v_{max,j}$) by taking the relative presence of mycorrhizae ($f_{cover,max}$) and their higher uptake efficiency ($f_{U_N^{eff}}^m$) compared to plants into account, while the adjustment of root biomass considers the disconnection of root tips from soil by mycorrhizal fungi. This coverage is given by the current ratio of fine root biomass ($C_f^p r$) and mycorrhizal biomass ($C^m$), but constrained by a maximum coverage to account for a shift in mycorrhizal community to more exploitative species, if plants are under severe N limitation.

In a similar way to plant C and N acquisition (eq. 1), mycorrhizal growth ($G^m$, fig. 2c,d) is defined as the net budget for C and N accumulation over time. $G_C^m$ is given by mycorrhizal C uptake from SOM ($U_C^m$, only in case of saprotrophic mycorrhizae), net C exchange between host plant and mycorrhizal fungi ($\Delta E_C$), and losses by mycorrhizal turnover ($T_C^m$), respiration ($R^m$), and exudation to accelerate SOM decomposition ($D_{SOM}^m$, only in case of decomposing mycorrhizae; fig. 2c). $G_N^m$ is given by mycorrhizal N uptake ($U_N^m$), net N exchange ($\Delta E_N$), and mycorrhizal turnover ($T_N^m$; fig. 2d).

$$\frac{dC^m}{dt} = G_C^m = U_C^m - \Delta E_C - T_C^m - R^m - D_{SOM}^m \tag{3a}$$

$$\frac{dN^m}{dt} = G_N^m = U_N^m - \Delta E_N - T_N^m \tag{3b}$$

The exchange of C and N between host plant and mycorrhizal fungi consists of an exudation flux from plants to mycorrhizae ($E^{p2m}$), where we assume that plant N exudation is zero, and the export flux from mycorrhizae to plants ($E^{m2p}$, fig. 3) is as follows:

$$\Delta E_C = +E_C^{m2p} - E_C^{p2m} \tag{4a}$$

$$\Delta E_N = +E_N^{m2p} \tag{4b}$$

whereby positive signs depict fluxes from mycorrhizae to plants and a negative signs depict fluxes from plants to mycorrhizae. We constrain C exudation ($E_C^{p2m}$) from the host plant to mycorrhizal fungi in a similar way as assumed in MYCOFON (Meyer et al. (2010)) by

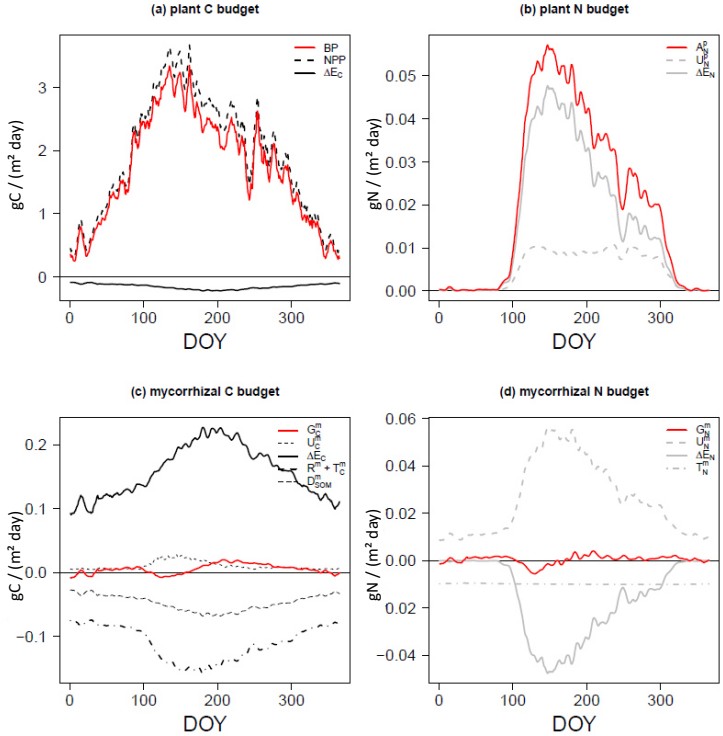

**Figure 2.** C and N budgets of plants (a,b) and mycorrhizal fungi (c,d) as described in equations 1 and 3. Net budgets in red, C fluxes in black, and N fluxes in grey. Uptake (U) in dashed lines, net exchange rates ($\Delta E$) in solid lines, losses by turnover (T) and respiration (R) in dashed-dotted lines (c,d), losses by exudation to SOM (D) in long dashes (c). Net exchange rates are partitioned into exudation from plant to mycorrhizal fungi ($E^{p2m}$; e, dashed) and export from mycorrhizal fungi to plant ($E^{m2p}$; e, dashed-dotted).

- the amount of available C, which is represented by the labile C pool in QUINCY

- plant N demand, where increasing N demand increases C allocation to mycorrhizal growth

- an interaction term, which allows for a decline in exudation, if plants do not receive nutrients in return, which could happen either in case of satisfied demand by plants, or in case of severe N limitation, which turns mycorrhizal fungi into competitors that do not deliver any N to host plants

- a minimum and maximum amount of mycorrhizae compared to plant fine roots, to avoid complete mycorrhizal death, and to prevent plants overspending C





**Figure 3.** C and N exchange rates between host plants and mycorrhizal fungi as described in equation 4. C fluxes in black, N fluxes in grey as in figure 2. Net exchange rates ($\Delta$E, solid lines) are partitioned into exudation from plant to mycorrhizal fungi (E$^{p2m}$, dashed) and export from mycorrhizal fungi to plant (E$^{m2p}$, dashed-dotted).

For a detailed description of the equations, see Appendix A.

We calculate N export from mycorrhizal fungi to host plants (E$_N^{m2p}$) as mycorrhizal surplus N, which is derived from current mycorrhizal N uptake (U$_N^m$) and N demand ($\zeta_N^m$), assuming that mycorrhizal fungi will first maintain their own C:N ratio ($\chi_{CN}^m$), before they can deliver N to host plants. This potential N export is scaled by plant N demand ($\zeta_N^p$) for actual N export to suppress export in case the plants have sufficient N. As we assume that mycorrhizal fungi deliver N as part of aminoacids, C export (E$_C^{m2p}$) is given by N export and a prescribed C:N ratio ($\chi_{CN}^{m2p}$).

$$E_N^{m2p} = (U_N^m - \zeta_N^m) \times \zeta_N^p, \; with \tag{5a}$$

$$\zeta_N^m = \frac{C^m}{\chi_{CN}^m} - N^m \tag{5b}$$

$$E_C^{m2p} = \chi_{CN}^{m2p} \times E_N^{m2p} \tag{5c}$$

Mycorrhizal N uptake (U$_N^m$) is the sum of mineral N uptake (U$_{N_j}^m$) and organic N uptake (U$_{N_{SOM}}^m$) in the case of saprotrophic mycorrhizae. Mineral N uptake, i.e. uptake of ammonium (NH$_4$) and nitrate (NO$_3$), is calculated similarly to plant root uptake in QUINCY (eq. 2), but down-regulated, if the C investment in N transformation to amino acids exceeds a threshold to avoid



mycorrhizal death:

$$U_{N_j}^{m,pot} = v_{max,N_j}^m \times N_j \times (K_{m1,N_j} + \frac{1}{K_{m2,N_j} + N_j}) \times C^m \times f_{invest}, \ where \tag{6a}$$

$$f_{invest} = \frac{f_{resp,max} \times C^m}{R_{U_{N_j}}^{m,pot}}, \tag{6b}$$

where $N_j$ denotes either $NH_4$ or $NO_3$. $v_{max,N_j}^m$ is a parameter describing the maximum uptake rate per unit mycorrhizal

biomass ($C^m$), and $K_{m1,N_j}$ and $K_{m2,N_j}$ are the low- and high-affinity half-saturation parameters. $f_{resp,max}$ is the empirical threshold for C investment into mineral N uptake.

We calculate the turnover of mycorrhizal fungi ($T_X^m$) based on the life span of mycorrhizae as:

$$T_X^m = \frac{1}{\tau_{myc}} \times X^m \tag{7}$$

where $\tau_{myc}$ is mycorrhizal turnover time and $X^m$ is either mycorrhizal C or N. $T_X^m$ is added to the respective litter pool in soil, similar to death plant fine roots.

We calculate respiration of mycorrhizal fungi ($R^m$) as the sum of direct mycorrhizal respiration ($R_{gm}^m$), which is associated with growth and maintenance processes, by a constant C-use efficiency ($CUE^m$) and recently gained C, and N uptake respiration

($R_{U_{N_j}}^m$), which is related to specific costs ($r_j$) to transform mineral N ($N_j$) into aminoacids (Zerihun et al. (1998)).

$$R^m = R_{mg}^m + R_{U_{N_j}}^m \tag{8a}$$

$$R_{mg}^m = (1 - CUE^m) \times (U_C^m + E_C^{p2m}) \tag{8b}$$

$$R_{U_{N_j}}^m = r_j \times U_{N_j}^m \tag{8c}$$

### 2.2.1 Special functionalities of saprotrophic and decomposing mycorrhizae

**Saprotrophic mycorrhizae** mine SOM as additional organic N source. We assume that mycorrhizae access old, C-depleted, N-rich SOM. We further assume that mycorrhizae are only facultative saprotrophs, which is why we decrease N uptake from SOM in case both plants and mycorrhizae have sufficient N, which is quantified by the current C:N ratio of mycorrhizae.

$$U_{N_{SOM}}^m = v_{max,N_{SOM}}^m \frac{N_{SOM}}{K_{m,N_{SOM}} + N_{SOM}} \times C^m \times f_{limit}, \ where \tag{9a}$$

$$f_{limit} = \begin{cases} 1, & if \frac{C^m}{N^m} > \chi_{CN,\frac{min}{2}}^m \\ 0.5, & if \chi_{CN,\frac{min}{2}}^m > \frac{C^m}{N^m} > \chi_{CN,min}^m, \ with \\ 0, & if \frac{C^m}{N^m} < \chi_{CN,min}^m \end{cases} \qquad \chi_{CN,\frac{min}{2}}^m = \frac{1}{2} \times (\chi_{CN}^m + \chi_{CN,min}^m) \tag{9b}$$

where $v_{max,N_{SOM}}^m$ is the maximum uptake rate per unit mycorrhizal biomass ($C^m$), and $K_{m,N_{SOM}}$ is the half-saturation parameter. $f_{limit}$ is a down-regulation function for N uptake from SOM, which is derived from the target C:N ratio of mycorrhizal





fungi ($\chi_{CN}^m$), a minimum C:N ratio ($\chi_{CN,min}^m$), and the current C:N ratio.

We assume that organic N uptake by SOM-mining mycorrhizae is associated with a C flux corresponding to the uptake of aminoacids and some additional C, even if most mycorrhizal fungi lack the capacity to assimilate soil organic C (SOC) and

acquire C only from their host plant ((Lindahl and Tunlid, 2015; Frey, 2019)). This C uptake ($U_C^m$) is described as:

$$U_C^m = U_{N_{SOM}}^m \times \chi_{CN}^{SOM} \times CUE^m \tag{10}$$

where $CUE^m$ is the C-use efficiency of mycorrhizal uptake, and describes the share of SOC that mycorrhizae can incorporate. The remaining C is respired.

**Decomposing mycorrhizae** exude C ($D_{SOM}^m$) to accelerate decomposition. This C provides energy for microbes to decompose older SOM faster, which enhances the net mineralisation flux, and therefore increases soil mineral N availability for mycorrhizal fungi, as well as plants and microbes. We calculate the exudation as:

$$D_{SOM}^m = f_{D_{SOM}} \times C^m \tag{11}$$

where $f_{D_{SOM}}$ is an empirical parameter. In order to not violate the fixed soil C:N ratio assumed in QUINCY, $D_{SOM}^m$ is added

to heterotrophic respiration. This is reasonable, as we assume that the induced acceleration of decomposition increases heterotrophic respiration generally.

We represent the resulting acceleration of decomposition implicitly by lowering the turnover time of slow-SOM in the presence of decomposing mycorrhizal fungi.

$$\tau_{SOM}^{*+} = \tau_{SOM}^* \times \frac{K_{m,\tau_{SOM}} + C^m}{v_{max,\tau_{SOM}} \times C^m} \tag{12}$$

where $\tau_{SOM}^*$ denotes the turnover time of the SOM without influence of decomposers, but in mycorrhizal presence. $K_{m,\tau_{SOM}}$, and $v_{max,\tau_{SOM}}$ are empirical parameters, and $C^m$ is mycorrhizal biomass.

In order to maintain overall ecosystem dynamics in steady state comparable to the model simulations without mycorrhizae,

we multiply the turnover of slow-overturning SOM pool in QUINCY by an empirically determined parameter $f_{\tau_{SOM}}$ for both mycorrhizal functionalities.

## 2.3   Duke Forest FACE experiment

We test the QUINCY-MYC model at the Duke Forest free-air $CO_2$ enrichment (FACE) site (hereafter Duke). The Duke FACE experiment (McCarthy et al. (2007)) was set up in a loblolly pine (*Pinus taeda*) plantation (35.9°N, 79.08°W, North Carolina,

US) that was established in 1983 after a clear-cut. The experiment started in August 1996 with three plots per $CO_2$ treatment, i.e. ambient and elevated (+200ppm) $CO_2$ concentration, with plots paired according to soil N availability.





**Table 1.** MYC model parameter

| symbol | description | value | unit | reference |
|---|---|---|---|---|
| $CUE^m$ | mycorrhizal C-use efficiency | 0.7 | $gC\ gC^{-1}$ | this study |
| $\epsilon$ | sensitivity parameter for plant C exudation | 0.005 | $\mu mol\ m^{-2} day^{-1}$ | this study |
| $f_{cover,max}$ | maximal covered root surface by mycorrhizal fungi | 0.3 | $m^2 m^{-2}$ | Meyer et al. (2010) |
| $f_{DSOM}$ | mycorrhizal C exudation to SOM per unit biomass | 0.05 | $gC\ gC^{-1} day^{-1}$ | this study |
| $f_{E_{C^{p2m}_{max}}}$ | maximal share of available C for exudation | 0.3 | $gC\ gC^{-1}$ | Meyer et al. (2010) |
| $f_{m2r,max}$ | maximum ratio of mycorrhizae to plant fine roots | 0.5 | $gC\ gC^{-1}$ | Marschner and Dell (1994); Göransson et al. (2006) |
| $f_{m2r,min}$ | minimum ratio of mycorrhizae to plant fine roots | 0.1 | $gC\ gC^{-1}$ | Marschner and Dell (1994); Göransson et al. (2006) |
| $f_{resp,max}$ | maximum C investment into N uptake | 0.5 | $gC\ gC^{-1} day^{-1}$ | this study |
| $f_{\tau_{SOM}}$ | SOM turnover adjustment | 5 | - | this study |
| $f^m_{U^{eff}_N}$ | surface ratio between plant fine roots and mycorrhizae | 40 | $m^2 m^{-2}$ | Marschner and Dell (1994) |
| $K_{m,N_{SOM}}$ | half-saturation parameter for $N_j$ uptake | 400 | $m^3 mol^{-1}$ | this study |
| $K_{m,\tau_{SOM}}$ | half-saturation parameter to $\tau_{SOM}$ reduction | 1.0 | $gC\ m^{-3}$ | this study |
| $\tau_{myc}$ | turnover time of mycorrhizae | 0.3 | years | Smith and Read (2010) |
| $v_{max,N_{SOM}}$ | maximum $N_{SOM}$ uptake capacity per unit biomass | 0.3 | $\mu mol\ mol^{-1} yr^{-1}$ | this study |
| $v_{max,\tau_{SOM}}$ | maximum $\tau_{SOM}$ reduction parameter | 20.0 | - | this study |
| $\chi^m_{CN}$ | C:N ratio of mycorrhizal fungi | 10.0 | $gC\ gN^{-1}$ | Allen et al. (2003); Wallander et al. (2003) |
| $\chi^m_{CN,min}$ | minimal C:N ratio of mycorrhizal fungi | $0.8 \times \chi^m_{CN}$ | $gC\ gN^{-1}$ | this study |
| $\chi^{m2p}_{CN}$ | C:N ratio of amino acids | 3.0 | $gC\ gN^{-1}$ | Hauptmann (1985) |
| $K_{m1,NH_4}$ | low affinity half-saturation parameter for $NH_4$ uptake | 0.0416 | $m^3 mol^{-1}$ | Thum et al. (2019) |
| $K_{m1,NO_3}$ | low affinity half-saturation parameter for $NO_3$ uptake | 0.0416 | $m^3 mol^{-1}$ | Thum et al. (2019) |
| $K_{m2,NH_4}$ | high affinity half-saturation parameter for $NH_4$ uptake | 1.0 | $m^3 mol^{-1}$ | Thum et al. (2019) |
| $K_{m2,NHO_3}$ | high affinity half-saturation parameter for $NO_3$ uptake | 1.0 | $m^3 mol^{-1}$ | Thum et al. (2019) |
| $r_{NH_4}$ | uptake respiration per unit $NH_4$ | 1.8 | $gC\ gN^{-1}$ | Thum et al. (2019) |
| $r_{NO_3}$ | uptake respiration per unit $NO_3$ | 2.3 | $gC\ gN^{-1}$ | Thum et al. (2019) |
| $\tau_{labile}$ | turnover time of plant labile pool | 7 | days | Thum et al. (2019) |
| $\tau_{SOM}$ | turnover time of slow SOM pool | 100 | years | Thum et al. (2019) |
| $v_{max,j}$ | maximum $N_j$ uptake capacity per unit biomass | 0.42 | $\mu mol\ mol^{-1} s^{-1}$ | Thum et al. (2019) |
| $\chi^{SOM}_{CN}$ | C:N ratio of SOM | 9.0 | $gC\ gN^{-1}$ | Thum et al. (2019) |



For our analysis, we use locally measured climate data (see below section model setup and protocol), annual measurement data for ambient and elevated treatments for plant C and N fluxes, i.e. plant biomass production (BP) and plant N acquisition ($A_N^p$), which are estimated by Finzi et al. (2007). Responses were calculated per each treatment plot pair and then averaged.

Soil C concentrations and $\delta^{13}$C measurements are taken from Lichter et al. (2008), who used the specific $\delta^{13}$C labelling of the artificial $CO_2$ that was used for fumigation.

## 2.4 Model setup and protocol

QUINCY requires half-hourly meteorological forcing consisting of air temperature, precipitation (rain and snow), longwave and shortwave radiation, atmospheric $CO_2$ concentration as well as N and P deposition rates. Meteorological data are derived

from the CRUNCEP dataset, version 7 (Viovy (2016)), which provides daily data from 1901 to 2015. Data are disaggregated using a statistical weather generator (Zaehle and Friend (2010)) to the half-hourly model time step. Annual data for atmospheric $CO_2$ concentration are taken from LeQuéré et al. (2018), and N deposition rates from Lamarque et al. (2010) and Lamarque et al. (2011). As local records of forcing data are available, we used data from Walker et al. (2014) for the duration of the Duke FACE experiment.

Additionally, QUINCY requires information about vegetation, given as plant functional type (PFT), and soil, such as texture, bulk density, and rooting and soil depth, for each site. As our later study aims for N controls on plant growth, we also use information about inorganic P content that is kept constant to avoid influence by P limitation.

QUINCY pools (vegetation and soil) are brought to quasi equilibrium by a spin-up period of 500 years that uses repeated

meteorological data from 1901 to 1930 to drive simulations, before the actual simulation period (1901-2015) starts with transient climate and $CO_2$ concentrations as described above. Vegetation tissue pools are set to zero in 1983, the year of forest establishment at the site. Harvested biomass persists in the system by adding it to litter, except for woody biomass, from which a fraction of 80% is removed.

We perform two simulations, one for ambient $CO_2$, and one where atmospheric $CO_2$ concentrations are elevated by 200ppm from August 1996 on, to mirror the FACE experiment. The $\delta^{13}$C signal is calculated by the amount of fumigated artificial $CO_2$ compared to natural $CO_2$ (McCarthy et al. (2007)).

To assess parameter uncertainty, we vary all MYC model parameters (tab. 1, above line) within a range of $\pm 10\%$. We use

latin hypercube sampling (LHS, Saltelli et al. (2000)) to generate 200 parameter sets for each mycorrhizal functionality. We did not vary default QUINCY parameters (tab. 1, below line), as a detailed analysis can be found in Thum et al. (2019). For the analysis of the Duke FACE experiment, we carry out two runs for each model variant and each parameter set, i.e. QUINCY without mycorrhizal fungi (one times two runs), QUINCY with decomposing mycorrhizae (200 times two runs), and QUINCY with saprotrophic mycorrhizae (200 times two runs). We analyse the median and the 10%- and 90%-quantiles of the





LHS simulations.

For better presenting changes to $CO_2$ elevation during the experiment, flux responses are shown as relative responses (eq. 13a, $\delta Y$), whereas changes in pool sizes, e.g. soil organic matter content or mycorrhizal biomass, are shown as absolute changes
(eq. 13b, $\Delta X$).

$$\delta Y = \frac{Y_e - Y_a}{Y_a} \tag{13a}$$

$$\Delta X = X_e - X_a \tag{13b}$$

In order to better understand the impact of the mycorrhizal parameterisation on the simulated ecosystem dynamics, we also use
a linear decomposition method (Rastetter et al. (1992)) to attribute the changes of ecosystem C storage under $eCO_2$ to changes in the N cycle induced by mycorrhizae.

## 3 Results

### 3.1 Effects on simulated vegetation dynamics under ambient and elevated $CO_2$

Simulated ambient plant biomass production (BP) is generally lower than observed plant biomass production ($999.6 \pm 138.8$
$gCm^{-2}yr^{-1}$)), but does not differ much among simulations with and without mycorrhizal fungi (fig. 4a). Simulated ambient plant N acquisition ($A_N^p$) captures observed plant N acquisition ($8.3 \pm 0.5$ $gNm^{-2}yr^{-1}$) in general (fig. 4c), whereby both plant biomass production and N acquisition are higher in simulations without mycorrhizae (BP: $808.7 \pm 39.5$ $gCm^{-2}yr^{-1}$; $A_N^p$: $9.0 \pm 0.2$ $gNm^{-2}yr^{-1}$) and with saprotrophic mycorrhizae (BP: $755.6 \pm 41.1$ $gCm^{-2}yr^{-1}$; $A_N^p$: $9.8 \pm 1.4$ $gNm^{-2}yr^{-1}$), and lower in simulations with decomposing mycorrhizae (BP: $678.1 \pm 71.2$ $gCm^{-2}yr^{-1}$; $A_N^p$: $6.3 \pm 0.5$ $gNm^{-2}yr^{-1}$). This indicates
a stronger competition for N by decomposing mycorrhizae than by saprotrophic mycorrhizae, which in turn limits plant growth. Root infection by mycorrhizal fungi lower direct plant N uptake, as they reduce the root contact to the soil ($U_N^p$, fig. 4e) by around 84%, however, this effect is balanced the by mycorrhizal N uptake and export to plants ($\Delta E_N$, fig. 4g). N export is higher for saprotrophic mycorrhizae ($8.4 \pm 1.4$ $gNm^{-2}yr^{-1}$) than for decomposing mycorrhizae ($4.5 \pm 0.4$ $gNm^{-2}yr^{-1}$), which leads to the simulated difference in total plant N acquisition.

Elevated $CO_2$ ($eCO_2$) increases both observed and modelled plant biomass production. The first-year effect is lower for observations (+24.8%) and simulations with saprotrophic mycorrhizae (+25.0%), and higher in simulations without mycorrhizae (+43.6%) and with decomposing mycorrhizae (+35.3%; $\delta BP$, fig. 4b). Observed plant biomass production response levels at $+27.2 \pm 2.1\%$ for the first six years of the experiment, before it increases to $+36.5 \pm 2.9\%$ for five years, and decreases
slightly in the last year (+20.3%). This behaviour is captured by none of the simulations entirely, but it also remains unclear whether the last-year decrease is a sign of progressive N limitation, caused by annual variation, or by a bias in observations





after winter 2001/2002, when a storm affected some of the plots (McCarthy et al. (2006)). Positive observed responses of plant N acquisition during the entire experiment phase (+23.5±6.2%; $\delta A_N^p$, fig. 4c) point to an effect by climate variability and disturbance rather than progressive N limitation. Modelled responses of plant biomass production are similar in simulations without mycorrhizal fungi and with saprotrophic mycorrhizae within the first half of the experiment. In the second half of the

experiment plant biomass production response simulated by the model without mycorrhizal fungi decreases, which is constrained by the small increase of simulated plant N acquisition within the first five years (+8.9±4.0%), and a slightly negative response afterwards (-5.0±5.5%). Contrary to that, simulated plant biomass production response by the model with saprotrophic mycorrhizae remains enhanced until the end of the experiment (+39.6±1.2%), which is supported by a strong positive response of plant N acquisition during the entire experiment (+33.8±12.9%). Simulated plant biomass production response by

the model variant with decomposing mycorrhizae is only positive during the first three years of the experiment, but slightly negative on average (-3.3±16.1%). This response is is caused by a small, but entirely negative response of plant N acquisition (-2.6±1.2%). Notably, this negative response of plant N acquisition is not caused by reduced plant N uptake in response to eCO$_2$ as in the model variant without mycorrhizal fungi ($\delta U_N^p$, fig. 4f). In the simulations, decomposing mycorrhizae accelerate N decomposition in response to eCO$_2$ and therefore allow plants to increase N uptake (+6.8±2.8%). However, they also act

as strong competitors, and while mycorrhizae take up N from the soil, they reduce N export to their host plants to satisfy their own N demand (-7.6±2.8%; $\delta \Delta E_N$, fig. 4h). In contrast to that, the simulated positive plant N acquisition response during the entire experiment in the model variant with saprotrophic mycorrhizae is caused by both a slight increase of plant N uptake (+1.8±5.7%) and strong increase in mycorrhizal N export (+40.3±14.9%), which matches observed total plant N acquisition response best.

**Table 2.** Change in mycorrhizal growth during the Duke FACE experiment.

| quantity | year | decomposers | saprotrophs |
|---|---|---|---|
| exudation response | 1996 | 0.6±0.4% | -3.2±5.9% |
| ($\delta E_C^{p2m}$) | 2000 | 1.6±2.2% | 2.2±17.8% |
| | 2005 | 8.5±0.8% | 8.1±5.3% |
| change in mycorrhizal biomass | 1996 | 19.3±6.3 gCm$^{-2}$ | 40.7±71.0 gCm$^{-2}$ |
| ($\Delta C^m$) | 2000 | 77.1±15.7 gCm$^{-2}$ | 192.6±113.9 gCm$^{-2}$ |
| | 2005 | 197.2±42.0 gCm$^{-2}$ | 197.7±62.3 gCm$^{-2}$ |

Differences in simulated N export rates from mycorrhizae to host plants ($\Delta E_N$) under ambient CO$_2$ among mycorrhizal types are caused by differences in total N uptake by mycorrhizae (U$_N^m$, fig. 5a). Generally, decomposing mycorrhizae take up almost 30% more N from mineral sources than saprotrophic mycorrhizae (U$_{N-min}^m$, fig. 5c; decomposers: 8.1±0.4 gNm$^{-2}$yr$^{-1}$, saprotrophs: 6.3±0.8 gNm$^{-2}$yr$^{-1}$). However accessing SOM as N source (U$_{N-SOM}^m$, fig. 5e; 7.2±1.5 gNm$^{-2}$yr$^{-1}$) allows saprotrophic mycorrhizae to more than double their N uptake, which leads to a 66% higher ambient mycorrhizal N uptake by

saprotrophic mycorrhizae than by decomposing mycorrhizae.



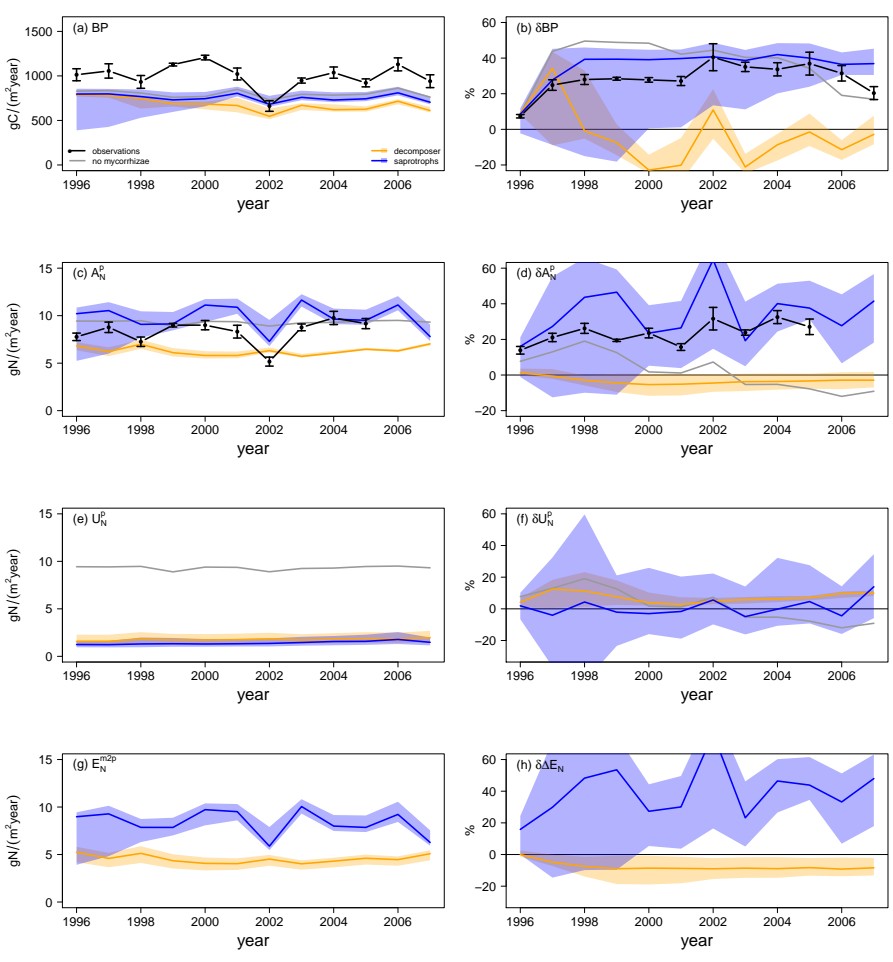

**Figure 4.** Vegetation C and N fluxes for the Duke FACE experiment for plant biomass production (BP; a,b), plant N acquisition ($A_N^p$; c,d), plant N uptake ($U_N^p$; e,f), and N export from mycorrhizae to plant ($E_N^{m2p}$; g,h), from ambient $CO_2$ simulations and treatments (a,c,e,g), and in response to elevated $CO_2$ (b,d,f,h), which is calculated by eq. 13a. Simulations: grey: QUINCY without mycorrhizae, orange: QUINCY with decomposing mycorrhizae, blue: QUINCY with saprotrophic mycorrhizae. Line represents LHS simulation median, and shaded areas present 10% - 90% quantile range of LHS simulations. Black: observations by Finzi et al. (2007), averaged over treatments with error bars indicating ±1 SE.



Under $eCO_2$ only saprotrophic mycorrhizae can increase their mycorrhizal N uptake significantly, by $+23.5\pm8.5\%$, whereas the simulated mycorrhizal N uptake response ($\delta U_N^m$) is almost zero for the model variant with decomposing mycorrhizae ($+1.5\pm1.5\%$; fig. 5b). This is caused by the greater competition for mineral N under $eCO_2$ by plants and decomposing mycor-

5   rhizal fungi, which constrains the $eCO_2$ response of mycorrhizal N uptake from mineral sources by decomposing mycorrhizae, and plant N uptake ($+4.9\pm1.6\%$) respectively (fig. 4f and fig. 5d). However, the slight increase of mineral N uptake in response to $eCO_2$ in comparison with the reduced N uptake in the model variant without mycorrhizal fungi, shows the effect of decomposing mycorrhizae on N cycling. Only the strong competition for mineral N then leads to the actually stronger limitation on plant growth. Contrary to that, saprotrophic mycorrhizae increase both uptake of mineral ($+5.5\pm5.8\%$) and organic N

10   ($+39.8\pm13.8\%$), in response to $eCO_2$ (fig. 5d,f), which results in the strong positive response of mycorrhizal N uptake in total that allows saprotrophic mycorrhizae to deliver more N to their host plants (fig. 4h). The positive response of mycorrhizal N uptake from mineral sources also indicates an increase in mycorrhizal biomass, especially in the beginning of the experiment, which is caused by an increased C exudation from plants to mycorrhizal fungi in response to $eCO_2$.

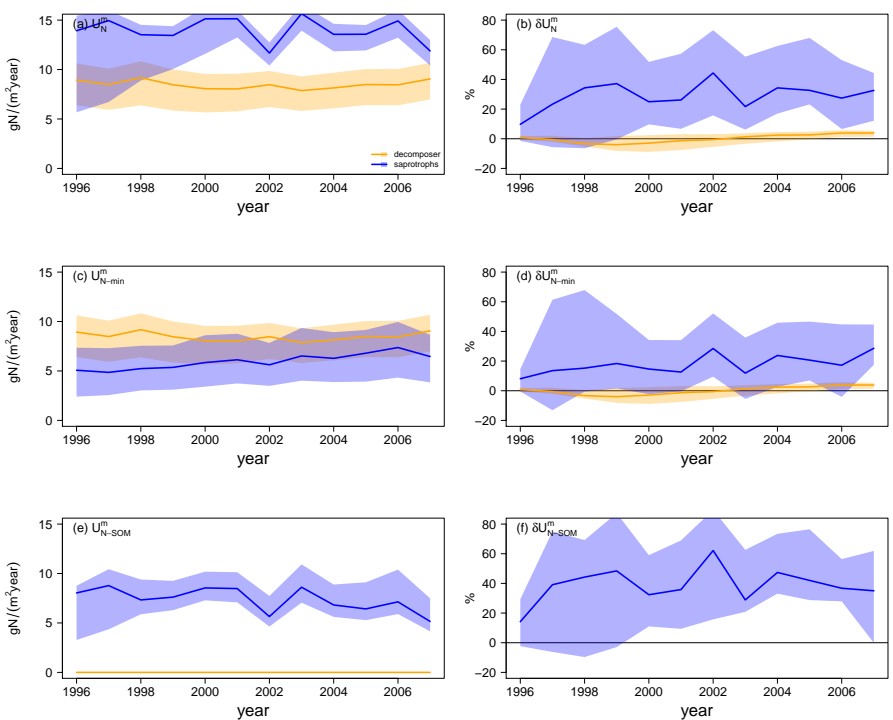

**Figure 5.** Mycorrhizal N uptake simulations for the Duke FACE experiment. Total uptake ($U_N^m$; a,b), uptake of mineral N ($U_{N-min}^m$; c,d), and N uptake from SOM ($U_{N-SOM}^m$; e,f), from ambient $CO_2$ simulations (a,c,e), and in response to elevated $CO_2$ (b,d,f), which is calculated by eq. 13a. Orange: QUINCY with decomposing mycorrhizae, blue: QUINCY with saprotrophic mycorrhizae. Line represents LHS simulation median, and shaded areas present 10% - 90% quantile range of LHS simulations.





Generally, the model behaviour does not change strongly by varying MYC model parameters by $\pm 10\%$ for both mycorrhizal types. For all analysed time series and responses to $eCO_2$ we find a larger spread among parameter sensitivity simulations for the saprotrophic mycorrhizae than for decomposing mycorrhizae (fig. 4 and fig. 5, shaded areas). The N-efficiency of exudation, i.e. the N gain per unit C that is exuded to mycorrhizal fungi, of saprotrophic mycorrhizae is strongly dependent on their

ability to mine N from SOM (eq. 9), which in the model is primarily controlled by the parameter $v^m_{max,N_{SOM}}$ that determines the maximum capacity of organic N uptake per unit mycorrhizal biomass. Increasing $v^m_{max,N_{SOM}}$ can increase the mycorrhizal surplus of N and thus the N export of saprotrophic mycorrhizae to plants. This creates a feedback in the model, in which increased return of N increases the amount of C that plants invest into exudation and subsequently also mycorrhizal biomass. Notably, the effect on plant C exudation can be positive, because plants support supportive mycorrhizal fungi, or negative,

in case mycorrhizal fungi export already enough N to fulfill plant requirements. Contrary to this, decomposing mycorrhizae always act as competitors for mineral N (fig. 4e and fig. 5c), so that plants exude the minimum amount of C required for their N demand. This implies that the modelled magnitude of decomposing mycorrhizae and their effect on plant N nutrition is insensitive to mycorrhizal parameters, but mostly controlled by the dynamically varying plant demand.

### 3.2 Effects on simulated soil dynamics under ambient and elevated $CO_2$

Mycorrhizal fungi also affect soil processes and soil organic C (SOC) stocks either by taking up and respiring C that was attached to soil organic N in case of saprotrophic mycorrhizae, or by accelerating SOM decomposition in case of priming mycorrhizae, which causes differences in the amount of SOC and its age during the FACE experiment. Thus, we also explore changes in total SOC, as well as the $\delta^{13}C$ signal of SOM. This gives an indication of how much of the SOC has been accumulated during the experiment in total, and also of the turnover of accumulated C, because a decrease in the $\delta^{13}C$ signal of SOM

indicates the respiration of younger C and thus potentially a faster ecosystem carbon turnover, as it carries the more negative $\delta^{13}C$ signal of artificial fumigation $CO_2$ (fig. 6a,b).



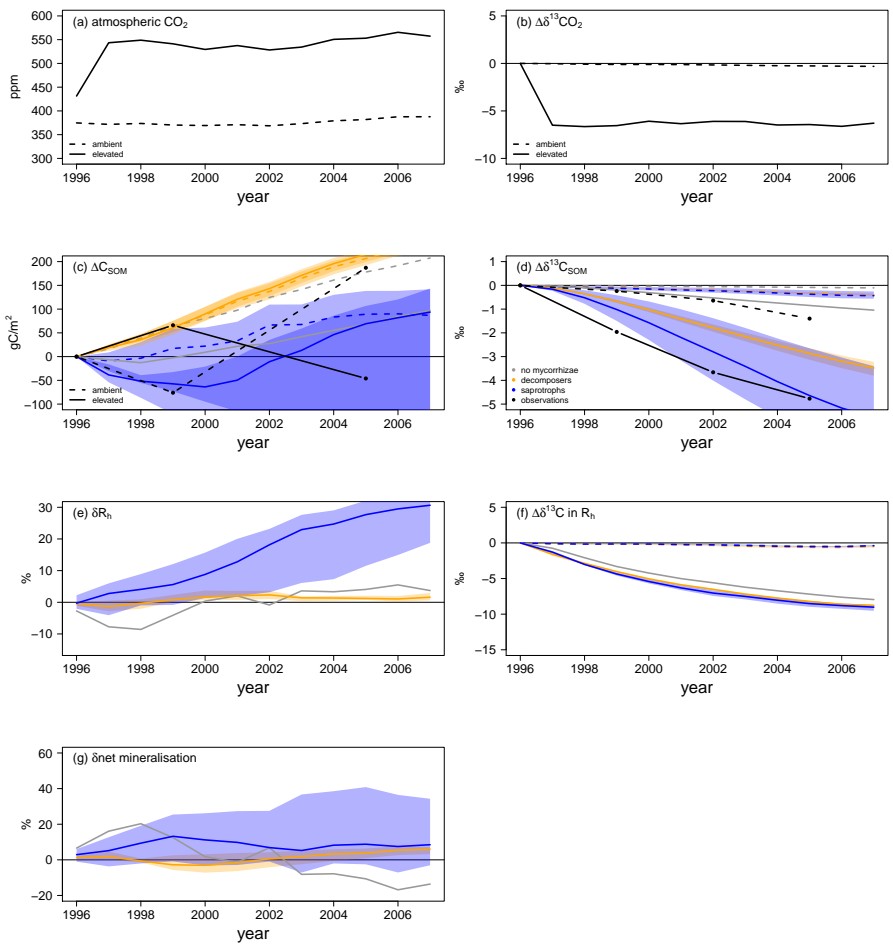

**Figure 6.** Soil C and N dynamics for the Duke FACE experiment. Subfigures a and b represent C forcing by atmospheric $CO_2$ concentration (a), and related $\delta^{13}C$ (b) for ambient and elevated $CO_2$ simulations. Subfigures c-g represent changes of soil dynamics related variables from the beginning of the experiment. SOC accumulation ($\Delta C_{SOM}$; c) and changes in SOM-$\delta^{13}C$ (d), responses of heterotrophic respiration and net mineralisation rate to elevated $CO_2$ ($\delta R_h$ and $\delta$net mineralisation, e,g), and changes in $R_h$-$\delta^{13}C$ (f). Absolute changes of $\delta^{13}C$ (b), SOC (c), SOM-$\delta^{13}C$ (d), and $R_h$-$\delta^{13}C$ (f) are calculated by eq. 13b, whereas relative changes of heterotrophic respiration (e) and net mineralisation rate (g) are calculated by eq. 13a. Subfigures a-d,f: ambient $CO_2$ treatment dashed and elevated $CO_2$ treatment solid. Subfigures c - g: simulations: grey: QUINCY without mycorrhizae, orange: QUINCY with decomposing mycorrhizae, blue: QUINCY with saprotrophic mycorrhizae. Line represents LHS simulation median, and shaded ribbons present 10% - 90% quantile range of LHS simulations. Black: observations by Lichter et al. (2008).

Observed ambient SOC ($SOC_a$) is increased by 187 $gCm^{-2}$ from 1996 to 2005, whereas SOC under $eCO_2$ ($SOC_e$) is decreased by 46 $gCm^{-2}$ (fig. 6c). Additionally $SOC_e$ has a more negative $\delta^{13}C$ signal (fig. 6d) indicating that even though less SOC is stored, more of this SOC has been created during the experiment. In the simulations, we find a similar pattern





for total SOC, i.e. a less strong SOC increase under $eCO_2$ than under $aCO_2$, in the model version without mycorrhizal fungi ($\Delta SOC_a$: +136 $gCm^{-2}$ in 2005, $\Delta SOC_e$: +38 $gCm^{-2}$), and in the version with saprotrophic mycorrhizae ($\Delta SOC_a$: +130 $gCm^{-2}$, $\Delta SOC_e$: +24 $gCm^{-2}$), even if the models still suggest accumulation of SOC over the experiment period contrary to observations (fig. 6c). However, SOC $\delta^{13}C$ for the model variant without mycorrhizal fungi shows only a very small response

to $eCO_2$ ($\Delta\delta^{13}C_a$: -0.1 ‰ in 2005, $\Delta\delta^{13}C_e$: -0.8 ‰), which indicates only minor changes in SOC turnover, whereas the $\delta^{13}C$ of SOC in the model variant with saprotrophic mycorrhizae becomes significantly negative ($\Delta\delta^{13}C_a$: -0.5 ‰, $\Delta\delta^{13}C_e$: -5.6 ‰). This indicates an accelerated turnover of SOC, which is induced by SOC mining and organic N uptake (fig. 6d). To the contrary, in the model variant with decomposing mycorrhizae, simulated total SOC evolution from 1996 to 2005 does not differ among $CO_2$ treatments ($\Delta SOC_a$: +174 $gCm^{-2}$, $\Delta SOC_e$: +178 $gCm^{-2}$). This is the consequence of a general tendency of

this model variant to allocate C below-ground by plant exudation to mycorrhizal fungi. Although mycorrhizal fungi accelerate SOM decomposition the effect is too small to prevent an increase in $SOC_a$. $SOC_e$ development is similar to $SOC_a$ development for the model variant with decomposing mycorrhizae, but the $\delta^{13}C$ signal is lower under $eCO_2$ than under $aCO_2$ ($\Delta\delta^{13}C_a$: -0.3 ‰, $\Delta\delta^{13}C_e$: -3.7 ‰). This shows again that freshly assimilated $CO_2$ is transferred to the soil via exudation and mycorrhizal fungi and that the turnover of C within the ecosystem becomes faster (fig. 6c,d).

Concurrent with the accelerated SOC turnover under $eCO_2$ during the experiment, heterotrophic respiration ($R_h$) is also increased in all model variants (fig. 6e). In the model variant without mycorrhizal fungi $\delta R_h$ becomes positive only after four years following the gradual accumulation of SOC (+2.9±1.7%). The similarity among SOC development under $aCO_2$ and $eCO_2$ by modelling decomposing mycorrhizae causes a weak response of $R_h$ (+1.0±0.8%, fig. 6e,g). However, the response

of net N mineralisation to $eCO_2$ increases slightly during the experiment (2.1±1.6%), suggesting that the decomposing functionality is accelerating soil turnover. While this change is too small to have a strong effect on the vegetation, it still leads to a notable amount of additional mineral N that is taken up by both plants (+4.9±1.6%, 4f) and mycorrhizal fungi (1.5±1.5%, fig. 5d). Simulated saprotrophic mycorrhizae access SOM as an N source and increase their organic N uptake in response to $eCO_2$ to satisfy their own and plant N demand (fig. 5f). The model assumes that the mycorrhizae only take up the C associated

with aminoacids, and respire the remainder of the SOC in order to maintain the SOM C:N ratio (eq. 10). This parameterisation leads to a strong increase of $R_h$ under $eCO_2$ (+12.3±10.1%; fig. 6e), associated with mycorrhizal uptake of N. The SOM decomposition unrelated to mycorrhizae is less affected, resulting in unchanged net N mineralisation (+0.5±3.8%; fig. 6g). This is similar to the model variant without any mycorrhizae (-1.3±10.6%). However, the $\delta^{13}C$ signal of $R_h$ is significantly lower when modelling mycorrhizal fungi, resulting from the larger below-ground C flux and faster turnover of mycorrhizal

biomass in the soil, in combination with the more negative $\delta^{13}C$ of the freshly assimilated C (fig. 6f).

These results are robust against parameter changes, but as discussed in the previous section model simulations show a larger spread among LHS simulations for saprotrophic mycorrhizae. This is due to the sensitivity of the MYC model with saprotrophic mycorrhizae to $v^m_{max,N_{SOM}}$, which describes the maximum C and N uptake from SOM (sec. 3.1), and thus directly

affects SOC turnover (eq. 10).



## 3.3 Effects on ecosystem C storage under ambient and elevated $CO_2$

The linear decomposition method after Rastetter et al. (1992) allows us to better understand the impact of the mycorrhizal parameterisation on the simulated ecosystem dynamics. We attribute the changes of ecosystem C storage under $eCO_2$ to changes in the N cycle induced by mycorrhizae (fig. 7), namely i) a change in total ecosystem N ($\Delta N_{Eco}$); ii) a change in the

C:N ratio of vegetation ($\Delta C{:}N_{Veg}$), iii) a change in the C:N ratio of SOM ($\Delta C{:}N_{SOM}$, including mycorrhizal fungi); and iv) an altered partitioning of N among ecosystem compartments, i.e. between vegetation and SOM ($\Delta N_{Veg}/N_{Eco}$).

We find that $\Delta N_{Eco}$ is only weakly changed under $eCO_2$ and has only minor effects on ecosystem C storage. This is because none of the model variants changes ecosystem N ($N_{Eco}$) gain, and effects on $N_{Eco}$ losses are minor.

For the model variant without mycorrhizal fungi, changes in ecosystem C are mainly caused by $\Delta C{:}N_{Veg}$, which indicates a

strong N limitation on plant growth, as increased carbon input from elevated $CO_2$ cannot be adequately matched by increased N uptake, and the models has insufficient capacity to increase the total below-ground carbon flux to reduce the excess C uptake (fig. 7). As a result, net C uptake response ($\delta NPP$) decreases after the initial increase (fig. 4b).

In the presence of mycorrhizal fungi, the lower attribution of $\Delta C{:}N_{Veg}$ to a change in ecosystem C under $eCO_2$ indicates less N limitation on growth. In the case of the decomposing mycorrhizae, small increases in decomposition due to increased

mycorrhizal growth (tab. 2) result in only minor changes in soil and vegetation C:N ratios, and are insufficient to lead to a substantial amount of N transfer from soils to vegetation, in summary leading to a small change in ecosystem C.

The improved N nutrition in the model variant with saprotrophic mycorrhizae that leads to an increase of ecosystem C is caused by a redistribution of N among ecosystem compartments, i.e. between vegetation and soil, due to the mycorrhizal mining of N from SOM and ensuing export of this N to vegetation ($\Delta N_{Veg}/N_{Eco}$, fig. 7). Since vegetation C:N ratios are more than three

times higher than soil C:N ratios (Kattge et al. (2011); Parton et al. (1993)), a redistribution of N from soil to vegetation allows vegetation to incorporate much more C additionally that exceeds the relative losses, which are caused by the loss of SOM-N.

As for vegetation and soil compartments individually (sec. 3.1 and sec. 3.2), the results are robust against parameter variations, which supports the meaning of our findings, especially when focusing on ecosystem responses to $eCO_2$.

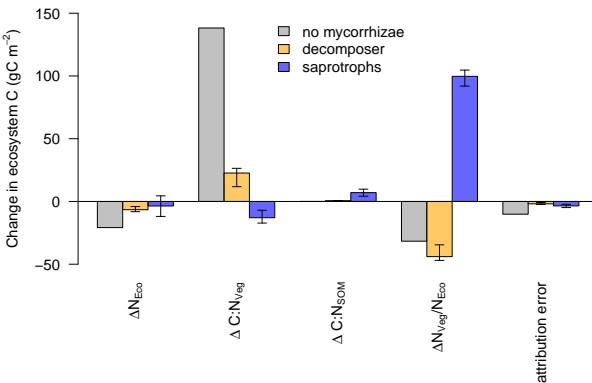

**Figure 7.** Attributions of the simulated change in ecosystem N ($\Delta N_{Eco}$), vegetation C:N ($\Delta C:N_{Veg}$), soil C:N ($\Delta C:N_{SOM}$), and ecosystem N distribution ($\Delta N_{Veg}/N_{Eco}$) to the simulated change in ecosystem C under elevated $CO_2$ for the model without mycorrhizal fungi (grey), with decomposing mycorrhizal fungi (orange), and with saprotrophic mycorrhizal fungi (blue). Bar heights indicate the median of the attributions, error bars indicate the 10% and 90% quantile spread among LHS simulations.

# 4   Discussion

## 4.1   Representation of alternative mycorrhizal functional groups

We integrated a dynamic plant-mycorrhiza-soil (MYC) model in the TBM QUINCY to test plausible but parsimonious representations of mycorrhizal effects on ecosystem fluxes based on observational findings. To do so, we chose incorporate observed mycorrhizal functionalities, but did not link them to mycorrhizal types, i.e. arbuscular mycorrhizae (AMs) and ectomycorrhizae (ECMs), and their potential ecosystem functioning, owing to contrasting observations and limited knowledge. This is contrary to other modelling studies, which differentiate between AMs and ECMs, and link their proposed effects on plant nutrition to the simulated host plants (e.g. Phillips et al. (2013); Brzostek et al. (2014); Sulman et al. (2014)). The advantage of our approach is that it allows us to assess potential generic effects of mycorrhizal dynamics on simulated ecosystems independently of plant species or any prior knowledge of species associations. We also tested the effect of mycorrhizal fungi that only access mineral N in soil (results not shown), which is similar to what plant fine roots do, but the results did not differ from the model version without mycorrhizae, indicating that mycorrhiza-SOM interactions are the key to explaining their significance for plant nutrient nutrition.

We assessed the possible importance of mycorrhizal fungi to predict the future evolution of terrestrial carbon stores under elevated $CO_2$. We found that only the model variant with saprotrophic mycorrhizae, in which mycorrhizae can directly access organic N bound in SOM, was able to simulate the observed growth response to $eCO_2$ and the underlying change in plant N acquisition consistent with observation-derived estimates by Finzi et al. (2007). The model without mycorrhizae was capable of simulating the observed increase in growth. However, this occurred only because of a too strong increase in the



C:N stoichiometry (Caldararu et al., 2020) and thereby nutrient-use efficiency of growth, comparable to simulations by other TBMs reported in Zaehle et al. (2014). This suggests that the direct access to SOM as N source is the key feature missing in TBMs when simulating Duke FACE. This result does not provide evidence for a general saprotrophic or at least SOM-mining behaviour of mycorrhizal fungi, but it is demonstrates that different N uptake strategies need to be explored to improve future

projections of ecosystem models. Our study indicates a strong need for both further modelling and experimental research on mycorrhizae.

The model variant in which mycorrhizae do not have direct access to organic N but induce an acceleration of SOM decomposition/priming was unable to reproduce either the observed growth or N acquisition response and predicted progressive N limitation. This result occurred because, on the simulated time scale, the implicit treatment of mycorrhizal effects on SOM

decomposition did not provide enough N for the plants, as mycorrhizal fungi, which are more efficient in taking up mineral N, hindered increased N export to plants as they required the additionally available N to support their own N demand. This turns the decomposing mycorrhizal fungi into competitors and not facilitators for N in the short-term, and consequently increased N limitation to plant growth despite the simulated acceleration of SOM decomposition and net N mineralisation under $eCO_2$. However, the set-up of FACE studies, especially their rapid $CO_2$ enhancement, is artificial, and the observation period

rather short. The results presented here may not represent the current environmental change as well as a full adjustment of the observed ecosystem to the change. On longer timescales the decomposing functionality may work to improve N nutrition of plants successfully.

Furthermore, our analysis clearly showed that the exudation of freshly assimilated C to any mycorrhizal fungi is necessary to explain the observed changes in the $\delta^{13}C$ signal in SOC (Lichter et al. (2008)). This shows that changes in soil C stocks can

only be simulated correctly when models include a form of C exudation to the soil.

## 4.2    MYC model simplifications and assumptions

### 4.2.1    MYC model limitations due to observational constraints

The MYC model contains 17 parameters in total. 13 of these belong to both functionalities, and control the plant-mycorrhiza interactions (e.g. C export to mycorrhiza) and general mycorrhizal dynamics (e.g. mycorrhizal stoichiometry and biomass

turnover). Two parameters each describe mycorrhizal effects on SOM specific for each functionality. We found observation-based estimates for only five parameters (tab. 1). This lack of suitable data is caused by large difficulties in measuring within the rhizosphere without destroying plant-mycorrhiza symbioses (Savage et al., 2018; Vicca et al., 2018; Lang et al., 2020), and obviously limits the ability to constrain the MYC model. Nevertheless, fluxes within the MYC model, which describe mycorrhizal C and N exchange with host plants and soil, are reasonably robust against changes in parameter values under

unperturbed conditions (fig. 4, fig. 5, and fig. 6). This provides some justification to circumvent the explicit representation of mycorrhizae and instead rely on the use of plant root measurements, i.e. plant fine-root biomass or plant C allocation into fine-root growth, as proxy for mycorrhizal biomass or C transfer from host plant to fungi. However, the resulting systematic underestimation of simulated flux responses to $eCO_2$ clearly points to the need to represent mycorrhizae and thus also the





observational data about mycorrhizal fungi. Our study reveals that in particular for the saprotrophic mycorrhizae, the organic N uptake capacity ($v_{max}^m$) needs to be better constrained. Even small variations of $v_{max,N_{SOM}}^m$ change the ability to export N to host plants, which feeds back to plant C exudation (fig. 3, eq. A3). Consequently, the MYC model is highly sensitive to $v_{max,N_{SOM}}^m$ when modelling saprotrophic mycorrhizae. The most important unknown parameters in the representation of

decomposing mycorrhizae are the mycorrhizal induced decomposition acceleration in combination with the general decelerated soil turnover ($f_{\tau_{SOM}}$ and $\tau_{SOM}^*$). As this does not affect plant-mycorrhiza interactions directly, the MYC model itself is less sensitive to variations, which leads to small variations among LHS simulations, but the combination strongly affect the entire ecosystem dynamics by potentially degrading almost all SOM.

### 4.2.2 MYC model limitations due to model constraints

In its current version, the MYC model does not take into account P and water supply by mycorrhizal fungi, and QUINCY neglects mycorrhizal biomass in plant leaf-to-root ratio calculations, which not only ensures a certain ratio of C to nutrient acquisition, but also plant physical stability. Consequently, QUINCY plants grow roots in order to fulfill physiological and physical requirements that also take up N and make mycorrhizae potentially redundant for plant N acquisition. The inclusion of mycorrhizal fungi to P and water supply, and plant leaf-to-root ratio calculations would therefore potentially intensify

the meaning of mycorrhizal fungi for host plants and probably reduce uncertainty, especially under changing environmental conditions.

The representation of the SOM as a pool with homogeneous C:N and chemical properties in QUINCY is a conceptual problem in most state-of-the-art TBMs. In reality, SOM is not a homogeneous pool, but a complex mixture of substrates with varying C:N ratios, microbial accessibility and decomposibility. Describing this variability with only two pools is a major lim-

itation and is particularly problematic for modelling SOM-mining mycorrhizae. Since in the model the C:N ratio of the slow SOM pool has to be maintained, uptake of organic N requires a loss of SOC either through respiration or incorporation into mycorrhizal biomass through the carbon contained in aminoacids as well as other substances. Even if saprotrophic behavior is occasionally observed (Wu et al. (2005), Treseder et al. (2007)), most mycorrhizae rely on plant C supply solely and lack saprotrophic abilities (Lindahl and Tunlid (2015); Frey (2019)). Using a more sophisticated representation of soil organic ma-

terial and its composition and dynamics using more physically based pool structures, stabilisation mechanisms and dependence of decomposition on microbial activity would be a pathway towards a more accurate and mechanistic modelling of organic N uptake from soils (Wutzler et al. (2017); Kyker-Snowman et al. (2020); Wang et al. (2020); Yu et al. (2020)).

### 4.3 Outlook: Do we need to incorporate mycorrhizal fungi into TBMs?

An increasing number of studies show that mycorrhizal fungi are an important factor for understanding vegetation-soil inter-

actions and soil dynamics, and predicting plant response to elevated $CO_2$. Our study underlines the importance of considering mycorrhizal fungi in ecosystem models to explain observed N acquisition plant responses under $eCO_2$. Additionally, we showed the importance of active plant C exudation to the rhizosphere in order to liberate N from SOM, which allows us to successfully reproduce observed changes in in SOC $\delta^{13}$C.

Generally, effects of mycorrhizal fungi on ambient simulations are minor, but the improved elevated $CO_2$ simulations, which suggests that our MYC model can better represent plant control on N acquisition than previous approaches under future conditions. A plant-controlled short-cut of the terrestrial N cycle by accessing SOM as nutrient source is important for reproducing observed responses to $eCO_2$, and thus indicates the necessity to incorporate the effects of mycorrhizal fungi on SOM decompo-

sition and plant nutrition into TBMs. However, the addition of any new process into the model always comes with the problems of added complexity, increased number of parameters and the possibility of equifinality. Especially given the little available data on mycorrhizae, their inclusion into TBMs can lead to increased uncertainty.

One possible solution can be an *implicit* formulation of mycorrhizal function in the form of active plant C investment into

rhizosphere processes that liberate N from SOM, commonly known as plant priming through root exudates. As we show that the interaction with SOM has the biggest effect on model predictions of ecosystem response under elevated $CO_2$, this may in itself improve the representation of plant N acquisition, and thus has the potential to reduce current uncertainty of modelled future land C uptake.

## 5    Conclusion

Here, we implemented a novel representation of mycorrhizal fungi into the terrestrial biosphere model QUINCY through representing mycorrhizal functionalities rather that taxonomic differences. We show that while the MYC model does little to improve predictions at ambient conditions it improves our predictions under elevated $CO_2$, most strikingly in the model variant which represents mycorrhizal interactions with soil organic matter. Our study highlights the importance of improving the representation of plant nutrient acquisition strategies in models to improve future predictions of plant growth responses

under increased atmospheric $CO_2$ levels, as well as the crucial data needed to parameterise and validate such models.

*Code availability.* The scientific part of the code is available under a GPL v3 licence. The scientific code of QUINCY relies on software infrastructure from the MPI-ESM environment, which is subject to the MPI-M-Software-License-Agreement in its most recent form (http://www.mpimet.mpg.de/en/science/models/license). The source code is available online (https://git.bgc-jena.mpg.de/quincy/quincy-model-releases), but its access is restricted to registered users. Readers interested in running the model should request a username and password from

the corresponding authors or via the git-repository. Model users are strongly encouraged to follow the fair-use policy stated on https://www.bgc-jena.mpg.de/bgi/index.php/Projects/QUINCYModel.

## Appendix A: Detailed model description

## A1    Plant C exudation to mycorrhizal fungi

C exudation ($E_C^{p2m}$) from host plant to mycorrhizal fungi is constrained by:





- the amount of available C, which is freshly assimilated C through photosynthesis (eq. A2)

- plant N demand, where increasing N demand increases C allocation to mycorrhizal growth (eq. A3a)

- an interaction term, which allows for a decline in exudation, if plants do not receive nutrients in return, which could happen either in case of satisfied demand by plants, or in case of severe N limitation, which turns mycorrhizal fungi into competitors that do not deliver any N to host plants (eq. A3b)

- a minimum and maximum amount of mycorrhizae compared to plant fine roots, to avoid complete mycorrhizal death, and to prevent that plants overspend C (tab. 1)

following the concepts of MYCOFON (Meyer et al. (2010)).

$$E_C^{p2m} = MIN(f_{m2r,min} \times C_{fr}^p - C^m, E_{C_{max}}^{p2m}) \tag{A1}$$

where $f_{m2r,min}$ is a parameter for the minimal amount of mycorrhizae compared to plant fine roots, $C_{fr}^p$ is plant fine root biomass $C^m$ is mycorrhizal biomass, and $E_{C_{max}}^{p2m}$ is the maximum exudation.

We determine the maximum exudation rate by available C, and the optimal ratio between mycorrhizae and fine roots under current conditions.

$$E_{C_{max}}^{p2m} = MIN(f_{E_{C_{max}}^{p2m}} \times C_{labile}^p \times \tau_{labile}, f_{m2r,opt} \times C_{fr}^p - C^m) \tag{A2}$$

where $f_{E_{C_{max}}^{p2m}}$ is the maximum share of available C that plants can allocate to mycorrhizal fungi, $C_{labile}^p \times \tau_{labile}$ describes the available C, and $f_{m2r,opt}$ is the optimal ratio between mycorrhizal biomass and plant fine root biomass.

We restrict the optimal ratio between mycorrhizal biomass and fine root biomass by a minimal and a maximal ratio, and scale it by plant N demand and mycorrhizal N support. Plant N demand, as well as mycorrhizal N support are used to represent current environmental conditions, which drive plant-mycorrhiza interactions, because plant N demand in depending on plant C uptake by photosynthesis that links to stand related conditions, such as plant type, successional stage, growing season, and atmospheric conditions, and on N availability that links to soil properties and that is also depicted by mycorrhizal N support.

$$f_{m2r,opt} = f_{m2r,min} + (f_{m2r,max} - f_{m2r,min}) \times \zeta_N^p \times \eta_N, \ with \tag{A3a}$$

$$\eta_N = \frac{E_N^{m2p}}{E_N^{m2p} + \epsilon} \tag{A3b}$$

where $f_{m2r,min}$ and $f_{m2r,max}$ are parameters, describing the minimal and maximal ratio between mycorrhizal biomass and





plant fine root biomass. $\zeta_N^p$, and $\eta_N$ are scaling factor between zero and one to describe plant N demand, and N support by mycorrhizae that is derived from mycorrhizal N export ($E_N^{m2p}$), and an infinitesimal small number ($\epsilon$) to stop C exudation, in case plants do not benefit from mycorrhizae over a time period of a month, and to technically ensure numerical stability.

5   *Author contributions.* MT and SZ designed the study and performed the analyses. MT developed the model. MT, SC and SZ interpreted the results. JE helped with model implementation. All authors contributed to writing the manuscript.

*Competing interests.* At least one of the (co-)authors is a member of the editorial board of Biogeosciences.

*Acknowledgements.* This work was supported by the European Research Council (ERC) under the European Union's Horizon 2020 research and innovation programme (QUINCY; grant no. 647204) and by the European Cooperation in Science & Technology (COST) within action
10   ES1308 (ClimMani; grant no. ES1308 - 39449).





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
