# Peer review of "Modelled forest ecosystem carbon-nitrogen dynamics with integrated mycorrhizal processes under elevated $CO_2$"

_Biogeosciences, 2023_

## Author Response (AR1)

Answer to Referee #1 by Josh Fisher

The reviewer comments are in bold, and the replies in regular font.

**This is a good paper, which represents among the leading edge of mycorrhizal incorporation into TBMs. By specifically not representing AM and ECM, but instead using mycorrhizal "functionalities", the authors present a different perspective on how mycorrhizae may be incorporated. The analysis is limited to only 1 site, though this site is a good one for testing these dynamics; but, given the limited spatial analysis the results and conclusions should be taken very loosely by readers.**

We thank Josh Fisher for his supportive and constructive review, and we agree that testing out model at only one site limits our analysis and conclusion. However, $CO_2$ experiments with this wealth of available data are rare, which is why we decided to use only one testing site. We tried to extend our analysis to multiple observation streams available at that site, i.e. aboveground biomass observations as well as soil C measurements. We have added a paragraph in the discussion to discuss this limitation and possible future extensions.

**Overall, I think this paper should be published in Biogeosciences. I have some questions that the authors should include as responses as well as text updates, in addition to some missed literature that the authors may find thought-provoking and useful for discussion within the paper.**

**Questions:**

- **What is the computational cost of including MYC?**

  Although the MYC model contains 15 equations, this is just a very minor part to the code-base of the QUINCY model, where most of the calculation time is allocated to the calculation of the energy, water balance and photosynthesis. Therefore, the added computational cost of MYC in QUINCY is very low and challenging to quantify even when running the full LHS in parallel for the FACE site (400 runs in total, 200 runs for each $CO_2$ treatment). Given the lack of relevance for the future use of MYC, we have not changed the manuscript with respect to this comment.

- **How can you parameterize this globally?**

  Data on mycorrhizal functionality is very limited locally but especially at global scale, so we agree with the reviewer that this is a critical aspect in scaling up our model. As an exploratory study, a next step could be to apply the model globally with default parameters to evaluate qualitatively the expected effects in scenarios with rising $CO_2$ (Braghiere et al. 2021). One argument for this capacity is that the LHS analysis demonstrated that the simulated effects were broadly robust against its parameterisation. However, we realise that a number of parameters will likely differ with PFT, soil type, climate etc. and therefore the quantitative result of such a model experiment would suffer from additional uncertainty given the lack of a suitable source of information for parameters and their global scaling (Braghiere et al., 2022). We will add a paragraph to discuss the challenges in scaling the findings up.

- **How does the model handle vegetation demographic (traits, PFTs, etc.) shifts e.g. with new bioclimatic envelopes with climate change and succession following disturbance, in relation to the mycorrhizal functions?**

  QUINCY in the version used in the study does not treat vegetation shifts or succession following disturbance, and is therefore not capable of evaluating these effects. The MYC model assumes that mycorrhizal fungi act more or less independent from the PFT and growth state of the host plants, as long as the plants exude sufficient amounts of carbon, which in the current form of the model does not vary across PFTs. We will revise the manuscript to clarify these limitations of the model for future study.

- **Interesting that with saptotrophs, the variability is captured better in Fig 4a-d. Show some scatterplots to highlight those aspects and provide more insight into the behavior?**

  Thank you for this comment, but we decided not to do that, because we wanted to focus on the causality, which we found was clearer in the shown timeseries than in scatter plots that potentially show a very good fit, but without any explanation. Additionally, since there are only annual measurements, a scatterplot would contain only a few plots.

**Minor notes:**

- **P21L4: strike "chose"**

  Thank you, this has been changed.

- **P23L29-30: refs.**

  We will add this to the text.

- **P24L9-10: this is what we did in FUN 1.0 (Fisher et al 2010), which created the framework for explicit incorporation of mycorrhizae in FUN 2.0 (Brzostek et al 2014).**

  Thank you for this comment. We will add this as reference.

**Additional literature:**

- **Incorporation of FUN 2.0 (Brzostek et al 2014, which you cite) into the larger TBM of CLM:**
  - **Shi, M., Fisher, J.B., Brzostek, E.R., Phillips. R.P., 2016. Carbon cost of plant nitrogen acquisition: global carbon cycle impact from an improved plant nitrogen cycle in the Community Land Model. *Global Change Biology* 22(3): 1299-1314.**
  - **Fisher, R.A., Wieder, W.R., Sanderson, B.M., Koven, C.D., Oleson, K.W., Xu, C., Fisher, J.B., Shi, M., Walker, A.P., Lawrence, D.M., 2019. Parametric controls on vegetation responses to biogeochemical forcing in the CLM5. *Journal of Advances in Modeling Earth Systems* 11(9): 2879-2895.**
  - **Lawrence, D.M., Fisher, R.A., Koven, C.D., Oleson, K.W., Swenson, S.C., Bonan, G., Collier, N., Ghimire, B., van Kampenhout, L., Kennedy, D., Kluzek, E., Lawrence, P.J., Li, F., Li, H., Lombardozzi, D., Riley, W.J., Sacks, W.J., Shi, M., Vertenstein, M., Wieder,**

W.R., Xu, C., Ali, A.A., Badger, A.M., Bisht, G., Brunke, M.A., Burns, S.P., Buzan, J., Clark, M., Craig, A., Dahlin, K., Drewniak, B., Fisher, J.B., Flanner, M., Fox, A.M., Gentine, P., Hoffman, F., Keppel-Aleks, G., Knox, R., Kumar, S., Lenaerts, J., Leung, L.R., Lipscomb, W.H., Lu, Y., Pandey, A., Pelletier, J.D., Perket, J., Randerson, J.T., Ricciuto, D.M., Sanderson, B.M., Slater, A., Subin, Z.M., Tang, J., Thomas, R.Q., Val Martin, M., Zeng, X., 2019. The Community Land Model version 5: Description of new features, benchmarking, and impact of forcing uncertainty. *Journal of Advances in Modeling Earth Systems*11(12): 4245-4287.

- **Update of FUN 3.0 with phosphorus, and into ELM:**
  - Allen, K.E., Fisher, J.B., Phillips. R.P., Powers, J.S., Brzostek, E.R., 2020. Modeling the carbon cost of plant nitrogen and phosphorus uptake across temperate and tropical forests. *Frontiers in Forests and Global Change*3(43): 1-12.
  - Braghiere, R.K., Fisher, J.B., Allen, K., Brzostek, E., Shi, M., Yang, X., Ricciuto, D.M., Fisher, R.A., Zhu, Q., Phillips, R.P, 2022. Modeling global carbon costs of plant nitrogen and phosphorus acquisition. *Journal of Advances in Modeling Earth Systems* 14(8): 1-23.
- **Discussion on global distributions of mycorrhizae and their impacts on global TBMs:**
  - Braghiere, R.K., Fisher, J.B., Fisher, R.A., Shi, M., Steidinger, B.S., Sulman, B.N., Soudzilovskaia, N.A., Yang, X., Liang, J., Peay, K.G., Crowther, T.W., Phillips, R.P., 2021. Mycorrhizal distributions impact global patterns of carbon and nutrient cycling. *Geophysical Research Letters*48(19): 1-11. e2021GL094514.

As mentioned before, we will add Braghiere et al., 2021 and Braghiere et al., 2022 to the text, because these are highly relevant to our paper.

**Good work!**

Thank you!

**Josh Fisher**

Answer to the Anonymous Referee #2

The reviewer comments are in bold, and the replies in regular font.

**This model description paper by Thurner et al. describe a new model formulation that explicitly considers the effect of mycorrhizal association in process-based ecosystem model QUINCY. The concept of splitting mycorrhizal associations into saprotrophic mycorrhizae and decomposing mycorrhizae is novel. The mathematical formulations are sound. The model evaluation against observations from the Duke FACE experiment is also thorough. I don't really any major concern over the manuscript, although I would love this see this model development applied at a P-limited site also, just to see how general the performance is. I think this paper would benefit the modelling community, and it certainly will stimulate the community to consider explicit representation of mycorrhizal association in land surface models.**

We thank the anonymous reviewer for this supportive review. We fully agree that applying the model at a P-limited site would add another insight into model performance. But at this development stage the model was only sensitive to N, and so QUINCY was run in its CN-only modus. Consequently, it would not respond to P limitation. Nevertheless, and as mycorrhizal fungi also support plant P acquisition, the model is ready its extension to P sensitivity, which will be done in a follow-up study. We will add a paragraph to the discussion to this effect.

**A suggestion which the author may or may not want to consider, is to run the model with both mycorrhizal types and see how it affects plant biomass production and N uptake and their CO2 responses.**

Originally, we tested the MYC model with four different types of mycorrhizae: mycorrhizae that did not interact with SOM (1), mycorrhizae that directly accessed SOM as N source (2, saprotrophs), mycorrhizae that indirectly interacted with SOM by exuding C to accelerate SOM decomposition (3, decomposers), and mycorrhizae that had both SOM-interaction abilities (4). We did not include types (1) and (4) into the manuscript for the following reasons:

Mycorrhizae (1) are only able to access inorganic N as N source, which is similar to plant N uptake. Thus, plants and mycorrhizae are competitors for inorganic N, whereby mycorrhizae have the better uptake abilities and cut plants' N access by their presence. In combination with the tight CN ratio of mycorrhizae plant N nutrition is worse than without mycorrhizae. So the presence of those mycorrhizae actually increases plant N limitation. This result just gave us the proof that the ability of mycorrhizae to interact with SOM is the necessary, but missing link in most land-surface models.

Mycorrhizae (4) always favor saprotrophic behavior over the decomposition acceleration in case they could do both for two reasons: saprotrophic behavior is the direct access of the N source, and thus the faster way to fulfill any N needs, and uptake of organic N from SOM, i.e. saprotrophic behavior, reduces SOM, which could be decomposed, which makes the decomposing ability even less effective in the presence of saprotrophic mycorrhizae. The only option to run both mycorrhizal functionalities at the same time would be to adjust parameters to lower the saprotrophic abilities in the presence of decomposing mycorrhizae. We decided not to do that, because we wanted to focus on both functionalities and their implications for plants and soil individually.

**Specific comments:**

**L8: through, not "though".**

Thank you, this has been changed.

**P7, Figure 2 caption: "Net exchange rates are partitioned into exudation from plant to mycorrhizal fungi (Ep2m; e, dashed) and export from mycorrhizal fungi to plant (Em2p; e, dashed-dotted)". What is "e" within the parentheses?**

That was a typo, which was deleted. Thank you!

**P8: Figure 3: DOY should be spelled in full.**

'day of year'' has been added to the figure caption.

**Equation 8a: "Rmg" is "Rgm" in L13. Make sure they are consistent.**

Thank you, this has been changed.

**P12, L34: are the two runs the ambient and elevated CO2 runs? Unclear in the text.**

We made two runs for each model version (and for each LHS parameter set), one under ambient CO2 and one under elevated CO2. We will make that clearer in the text.

**P14, L2: experimental, not "experiment". Also, do you have a reference to support this?**

Thank you, this has been changed and we will look for a reference.

**P14, L4 – 7: This is a long sentence, and the message is not very clear. I suggest to break it into two sentences.**

Yes, thank you, corrected.

**L4, L11: two "is".**

Thank you, this has been changed.

**Figure 6: Do you have error bars for the observations, especially for soil C?**

We took the values from Lichter et al., 2008 (tab. 3), but we show changes during the experiment instead of absolute values to avoid too strong effects by initial value differences among model versions with/without mycorrhizal fungi. Changes in SOC/d13C during the experiment have about the same size or are even smaller than the given standard deviations of the measurements, which is why we decided not to present them in our figure, but to focus on the average pattern.

**P18, last sentence to P19, L4. This is a very long sentence. Consider rephrase it.**

Yes, thank you, we try to rephrase it.